# Identification of Metabolic Factors and Inflammatory Markers Predictive of Outcome after Total Knee Arthroplasty in Patients with Knee Osteoarthritis: A Systematic Review

**DOI:** 10.3390/ijerph20105796

**Published:** 2023-05-11

**Authors:** Lotte Meert, Michel GCAM Mertens, Mira Meeus, Sophie Vervullens, Isabel Baert, David Beckwée, Peter Verdonk, Rob J. E. M. Smeets

**Affiliations:** 1Research Group MOVANT, Department of Rehabilitation Sciences and Physiotherapy (REVAKI), University of Antwerp, 2610 Wilrijk, Belgium; 2Pain in Motion, International Research Group, Vrije Universiteit Brussel, 1050 Brussels, Belgium; 3Department of Rehabilitation Sciences, Ghent University, 9000 Ghent, Belgium; 4Rehabilitation Research (RERE) Research Group, Department of Physiotherapy, Human Physiology and Anatomy, Faculty of Physical Education & Physiotherapy (KIMA), Vrije Universiteit Brussel, Laarbeeklaan 103, 1090 Brussels, Belgium; 5Antwerp Orthopaedic Center, AZ Monica Hospitals, 2018 Antwerp, Belgium; 6Department of Rehabilitation Medicine, Research School CAPHRI, Maastricht University and CIR Revalidatie, 5628 WB Eindhoven, The Netherlands

**Keywords:** knee osteoarthritis, total knee arthroplasty, systematic review, postoperative outcome, metabolic factors, low-grade inflammation, chronic pain

## Abstract

Objective: To identify metabolic factors and inflammatory markers that are predictive of postoperative total knee arthroplasty (TKA) outcome. Method: A systematic search of the existing literature was performed using the electronic databases PubMed, Web of Science and Embase until the 1^st^ of August 2022. Studies that evaluated the influence of metabolic or inflammatory markers (I) on postsurgical outcome (O) in end-stage knee osteoarthritis patients awaiting primary TKA (P) were included in this review. Results: In total, 49 studies were included. Risk of bias of the included studies was low for one study, moderate for 10 studies and high for the remaining 38 studies. Conflicting evidence was found for the influence of body mass index, diabetes, cytokine levels and dyslipidaemia on pain, function, satisfaction and quality of life at more than six months after TKA. Conclusions: Several limitations such as not taking into account known confounding factors, the use of many different outcome measures and a widely varying follow-up period made it challenging to draw firm conclusions and clinical implications. Therefore large-scaled longitudinal studies assessing the predictive value of metabolic and inflammatory factors pre-surgery in addition to the already evidenced risk factors with follow-up of one year after TKA are warranted.

## 1. Introduction

About 20% of patients undergoing TKA report unfavourable outcomes after surgery, such as chronic postsurgical pain [1,2]. This pain has a major impact on patients and is often associated with functional deficits, worse general health, anxiety, depression, sleep problems and long-term opioid use [3,4]. Unfortunately, it is mostly very difficult to identify the aetiology of chronic pain after TKA, resulting in a significant subset of patients with unexplained persistent pain [5].

Determining risk factors associated with poor postoperative outcome after TKA is of tremendous importance to identify patients at risk. This could result in more accurate patient selection and creating more realistic expectations [6]. Previous systematic reviews have found evidence for a range of modifiable and non-modifiable patient-related preoperative risk factors including female gender, lower age, low socio-economic status, more preoperative pain, comorbidities and psychological factors (depression, anxiety or pain catastrophizing) [7].

Besides these already known risk factors, metabolic factors and inflammatory markers might also influence postoperative TKA outcome. Metabolic disorders such as obesity, diabetes and, by extension, the metabolic syndrome (which is the commonly observed clustering of obesity, hypertension, dyslipidaemia, and insulin resistance) are risk factors for developing knee OA in the first place [8,9,10], but if and how they relate to the development of postoperative chronic pain is still unclear. Recently, it has become clear that OA is more than a “wear-and-tear” disease and that the presence of metabolic factors and inflammatory markers such as C-reactive protein, pro-inflammatory cytokines and chemokines, also playing an important role in diabetes and obesity and the metabolic syndrome [11], cannot be underestimated in the development of OA but also in the modulation of pain processes (24). Research shows that increased systemic inflammation is associated with higher preoperative patient-reported pain levels in patients with knee OA [12]. In addition, systemic inflammation is characterized by high levels of circulating proinflammatory cytokines (such as interleukin-6 (IL-6) and tumour necrosis factor-α (TNFα)), which can sensitize the peripheral and the central nervous system [13,14]. This altered central pain processing may play an essential role in the development and maintenance of chronic pain. Besides being common in patients with OA [15,16], preliminary research suggests that altered central pain processing may even be a determinant of lower long-term benefit from joint replacement surgery. Hence, metabolic factors and inflammatory markers might be related to postoperative outcome.

A better understanding of how metabolic disorders, metabolic factors and inflammatory markers are related to postsurgical TKA outcome might increase our insight into the timing of surgery, patient expectations, patient-surgeon shared decisions and preoperative treatment decisions. Therefore, the aim of this study is to systematically review and critically appraise the existing evidence related to metabolic factors and inflammatory markers predictive of pain, functional disabilities, quality of life (QoL) and patient satisfaction after TKA in patients with knee OA.

## 2. Methods

### 2.1. Protocol and Registration

This systematic review is reported following the PRISMA-guidelines (Preferred Reporting Items for Systematic reviews and Meta-Analyses) [17]. This review protocol was prospectively registered at PROSPERO (registration number CRD42022350609).

### 2.2. Eligibility Criteria

To be included in this review, articles had to report results of studies that evaluated the influence of preoperative metabolic factors or inflammatory markers (I) on postoperative pain, functional disabilities, quality of life or patient satisfaction (O) in end-stage knee OA patients awaiting TKA (P). Full inclusion and exclusion criteria can be found in Table 1.

### 2.3. Information Sources and Search

A systematic search of the existing literature was performed on the 1 August 2022, using the electronic databases PubMed, Web Of Science and Embase. The search strategy was based on three groups of search terms related to “Knee OA and Total Knee Replacement Surgery (P)”, “Metabolic Factors or inflammatory Markers (I)” and “Postsurgical Outcome (O)”. The construct of the search strategy is available as Appendix A.

### 2.4. Study Selection

Eligibility assessment was performed independently in a blinded standardized manner by the first and the second author (L.M. and M.M.) using Rayyan [18]. First, all search results were screened based on title and abstract. The full-text article was retrieved if the citation was considered potentially eligible and relevant. In the second phase, each full-text article was again evaluated whether it fulfilled all criteria. If any of the eligibility criteria was not fulfilled, the article was excluded. Disagreements between reviewers were resolved by consensus. If there were any disagreements after discussion, the opinion of the third author (M.M.) was provided.

### 2.5. Data Collection Process and Data Items

A data extraction sheet was developed and completed by the first author (L.M.). Following data was extracted from the included articles: (1) author, year of publication and study design; (2) characteristics of the study population; (3) specification of the examined preoperative metabolic and/or inflammatory factors; (4) timing of measurements; (5) primary postoperative outcome measures; (6) statistical analysis used (univariate or multivariate); (7) key findings related to the influence of preoperative metabolic and/or inflammatory factors on postoperative outcome(s).

### 2.6. Risk of Bias (RoB)

The full-text versions of all studies that met the inclusion criteria were retrieved for assessment of RoB using the Quality in Prognosis Studies (QUIPS) tool developed by Hayden et al. (28). This was done by the first author (L.M.). The QUIPS considers six domains of potential biases: (1) study participation; (2) study attrition; (3) prognostic factor measurement; (4) outcome; (5) measurement of and controlling for confounding variables; and (6) analysis approaches. Each criterion was answered using “yes” (criterion fulfilled), “no” (criterion not fulfilled), or “unclear”. For each of the 6 potential biases, a study was rated as having low, moderate, or high RoB per domain. A study was rated as low RoB if all domains were at low RoB or up to one was moderate RoB. A study was scored as moderate RoB when there were at least two domains at moderate, but not at high RoB in any domain. A high risk was judged when at least one domain was at high risk.

Further, levels of evidence of studies were determined with the Evidence Based Guideline Development (EBRO) approach, an initiative of the Dutch Cochrane Center and the Dutch Institute for Healthcare Improvement [19]. In accordance with this methodology, selected studies were classified according to their methodological quality and strength of evidence: A1: systematic review including at least two independent A2 level studies; A2: prospective cohort study of substantial size and sufficiently long follow-up period, adequate control of confounders and minimal chance of selective drop-out during follow-up; B: prospective cohort study, but not having all characteristics of an A2 study, or a retrospective cohort study or case-controlled trial; C: non-comparative study; and D: expert opinion. Finally, levels of conclusion are determined according to the EBRO method [19]. Level 1 evidence is represented by one A1 study or at least two independent A2 studies. Level 2 evidence is represented by one A2 or at least two independent B studies and Level 3 evidence is represented by one B or C study or conflicting results. Finally, level 4 evidence is represented by expert opinion only.

## 3. Results

### 3.1. Study Selection and Characteristics

The literature search identified 3.249 studies for screening of which 49 were included in this systematic review (Figure 1). The most important reasons for exclusion were inappropriate study design (e.g., retrospective study), inappropriate postoperative outcome (e.g., follow-up <6 months, outcome other than pain, functional disabilities, QoL, or satisfaction), and non-eligible population (patients with rheumatoid arthritis or unicompartimental knee arthroplasty). Of the 49 included studies, two were RCTs [20,21], two case-control studies [22,23] and 45 prospective observational studies [24,25,26,27,28,29,30,31,32,33,34,35,36,37,38,39,40,41,42,43,44,45,46,47,48,49,50,51,52,53,54,55,56,57,58,59,60,61,62,63,64,65,66,67,68] (of which two were secondary analyses [61,62]). For each study, the characteristics are presented in Table 2. The number of patients ranged from 28 (29) to 11.084 (30) and the follow-up period after surgery ranged from six months [22,24,27,29,30,33,42,43,45,52,55,57,65,67] to 17 years [39].

### 3.2. Risk of Bias Assessment

The RoB of the reviewed studies is presented in Table 3.

The overall RoB was low for one study [62], moderate for 10 studies [21,23,29,34,36,38,41,46,52,56], and 38 studies suffered a high RoB [20,22,24,25,26,27,28,30,31,32,33,35,37,39,40,42,43,44,45,47,48,49,50,51,53,54,55,57,58,59,60,61,63,64,65,66,67,68]. RoB was mainly due to lack of information about study attrition and confounding factors. Other reasons were the lack of information or use of obvious valid and reliable prognostic factor measurements. Either no information regarding measurement of the prognostic factor was given or the information was retrieved from patient records. One study [62] had level A2 of evidence and all other studies were at level B of evidence [20,21,22,23,24,25,26,27,28,29,30,31,32,33,34,35,36,37,38,39,40,41,42,43,44,45,46,47,48,49,50,51,52,53,54,55,56,57,58,59,60,61,63,64,65,66,67,68].

### 3.3. Outcome Measures

An overview of the outcome measures that are used to assess pain, function, QoL and patient satisfaction can be found in Table 4. Function is further divided into impairments and activities/limitations, and gait related impairments and gait related activities/limitations. Gait related outcome measures will be discussed separately in order to keep a clear overview and to summarize findings based on more homogenous outcomes.

Studies reporting change in outcome measures (preoperative vs. postoperative results) will be reported separately from studies reporting absolute postoperative values of the outcome measures.

### 3.4. Predictive Factors

A summary of the results for the univariate and multivariate analyses can be found in Table 5. The results are described below for each factor and each of the outcome measures.

#### 3.4.1. Factor: Body Mass Index

##### Pain

*Relative change in outcome*: Two univariate analyses [26,45] and one multivariate analysis [43] showed more pain reduction from baseline to postoperative outcome at six months [43,45] and at one year [26], in favour of obese patients. In contrast, the univariate analysis of Mishra et al. reported less pain reduction in obese patients [65]. In contrast, another multivariate analysis showed similar improvement in pain between non-obese and obese patients two years postoperatively [29].

*Absolute outcome*: Two univariate [23,44] and four multivariate analyses [29,43,48,52] found that higher BMI was not predictive for more pain six months [43,52], one year [23,48], two years [29] and five years [44] postoperatively. In contrast, one multivariate analysis found that class III obesity was associated with more pain three years postoperatively [47].

##### Functional Impairment

Four studies examined the influence of BMI on knee ROM [21,40,52,68].

*Relative change in outcome*: One multivariate analysis reported greater knee ROM improvement in patients with higher BMI [68].

*Absolute outcome*: One univariate analysis showed that knee ROM in obese patients was significantly lower than in non-obese patients one year and 10.8 years postoperatively [40]. However, the multivariate analysis of Pua et al. did not find BMI to be a predictor of postoperative knee ROM six months postoperatively [52]. In another study of Lampe and colleagues, lower BMI in combination with higher preoperative knee flexion predicted a higher maximal knee flexion one year postoperatively [21].

Two univariate [30,40] and five multivariate studies [32,33,41,44,68] examined the influence of BMI on the KSS knee outcome score.

*Relative change in outcome*: One study found lower BMI to be associated with better KSS knee outcome scores at nine years [30] contrary to Zhang et al. who found that BMI was not a predictor two years postoperative [68].

*Absolute outcome*: While one study found lower BMI to be associated with better KSS knee outcome scores 10.8 years [40] postoperatively, four other studies stated that BMI was not a predictor for KSS knee outcome scores at six months [33], at one year [32,33], at two years [33] and at five years [41,44] follow-up.

Two univariate [23,45] and one multivariate [47] studies examined the influence of BMI on WOMAC stiffness.

*Relative change in outcome*: One univariate analysis found that obese patients showed better WOMAC stiffness scores six months postoperatively [45] while another univariate [23]

*Absolute outcome*: The multivariate analysis of Nunez et al. showed no effect of BMI on WOMAC stiffness [47].

##### Functional Activities/Limitations

*Relative change in outcome*: Three univariate [26,27,59] and two multivariate analysis [29,68] showed a similar gain in function between obese and non-obese patients six months [27], one year [26], two years [29,68] and five years [59] postoperatively. However, Mc Queen et al. (univariate) and Zhang et al. (multivariate) found that obese patients showed more functional gain six months [45] and two years [68] postoperatively and contrary to this, two other univariate analyses found larger functional improvement in non-obese patients 10.8 year [39] and one year [65] postoperatively.

*Absolute outcome*: Eleven studies, of which one conducted a univariate analysis [23] and 10 multivariate analyses [25,29,32,33,34,44,47,56,61,62], found that BMI was not a predictor for functional outcome at six months [29,33], 1 year [23,32,33,56,61], two years [29,33], three years [34,47] and five years [25,44] postoperatively or for sedentary behaviour one year postoperatively [62]. Thirteen studies, of which five were univariate analyses [30,37,40,44,59] and eight multivariate analyses [20,22,35,41,48,50,55,62] found that non-obese patients had significantly better function than the obese group at six months post-surgery on KOOS ADL and KOS-ADLS [22,55], at one year on WOMAC, OKS, KSS function, reported physical activity [35,37,40,50,62], at five years on KSS [41,59] at nine years on KSS function [30] and at 10.8 years on WOMAC [40].

##### Gait Impairments

*Relative change in outcome*: One multivariate analysis found no association between BMI and knee ROM during the gait cycle [26] and Paterson et al. found that gait biomechanics were not influenced by BMI, two years postoperatively [49].

##### Gait Activities/Limitations

Six multivariate analyses [22,26,42,52,55,60] looked at the influence of BMI on gait related activities/limitations.

*Relative change in outcome*: One study did not show a significant association between BMI and gait velocity gain one year postoperatively [26].

*Absolute outcome*: Two other studies confirmed that postoperative walking speed 6 months postoperatively was not predicted by preoperative BMI [42,55]. Pua et al. found that BMI was not predictive of walking limitations at six months post-surgery [52]. In addition, stair climbing speed and TUG at six months postoperative were not influenced by BMI [55], idem for handrail use during stair climbing two years postoperatively [60]. The opposite was found in two other studies where lower BMI predicted better gait outcomes (gait speed and steps/day) [22] and a better stair climbing speed [42] at six months post-surgery.

##### Satisfaction

*Absolute outcome*: Four multivariate analyses showed that BMI was not predictive of satisfaction at one year [53,61], two years [29] and five years [63] post-surgery. Two other studies, one conducting univariate analyses [37] and one multivariate analyses [46], showed that less obese patients were more satisfied one year [37] and two years postoperatively [46].

##### Quality of Life

*Relative change in outcome*: Eight studies, of which six performed univariate analyses [26,27,31,45,59,65] and only two multivariate analyses [43,68], looked at the role of BMI on gain in QoL. Three studies (including the multivariate analysis) showed a similar gain in QoL between obese and non-obese patients at six months [43], at one year [26] and at five years [59]. Further, two studies showed a larger increase in QoL in favour of non-obese patients one year postoperatively [27], while in contrast two other studies showed a significantly larger QoL gain in favour of obese patients at six months [45] and at one year [31,65]. One multivariate analysis reported no difference in improvement for the Short Form 36 mental composite score (SF-36 MCS) but greater improvement of the Short Form 36 physical composite score (SF-36 PCS) in favour of higher BMI two years postoperative [68].

*Absolute outcome*: In addition one univariate [44] and three multivariate analyses [34,55,56] showed that BMI was not a predictive factor of QoL six months [55], one year [56], three years [34] and five years post-operatively [44]. In contrast, four other studies (two univariate [37,59] and two multivariate analyses [43,54]) found that greater level of obesity resulted in worse QoL at six months [43], one year [37], two years [54] and at five years [59].

##### Conclusion

Concerning the influence of BMI on postoperative pain, functional impairments, functional activities/limitations and QoL, both univariate and multivariate studies reported conflicting results. Some results were in favour of obese patients meaning that more obese patients showed less pain, less functional impairments, less functional limitations and better QoL after TKA. Some in favour of non-obese patients meaning that non-obese patients showed less pain, less functional impairments, less functional limitations and better QoL after TKA and some were similar for obese and non-obese patients. Regarding the influence of BMI on gait impairments, gait activities and satisfaction, results were also conflicting, however results were never in favour of obese patients meaning that more obese patients did never show better gait functions or were never more satisfied than less obese patients.

#### 3.4.2. Factor: Diabetes

##### Pain

*Relative change in outcome*: In one multivariate analysis, diabetes that impacted routine activities showed less pain reduction at six months [24]. While another multivariate analyses showed that diabetes was and at one year [64] postoperative.

*Absolute outcome*: According to Pua et al. diabetes was not predictive for pain at six months [52].

##### Functional Impairment

*Relative change in outcome*: One multivariate analysis reported that diabetes resulted in poorer improvement two years postoperatively [68].

*Absolute outcome*: According to one univariate analysis, diabetes was not predictive of knee ROM two years postoperatively [58], while two multivariate analyses found that absence of diabetes was predictive of better knee ROM at six months [52] and at two years postoperatively [68]. One multivariate analysis found no association between diabetes and postoperative knee function at five years [41].

##### Functional Activities/Limitations

*Relative change in outcome*: One multivariate analysis also reported less functional improvement at six months in patients with diabetes that impacts routine activities [24]. In contrast, three multivariate analyses found that diabetes did not influence postoperative function at one year [50,64] and at two years [68].

*Absolute outcome*: One univariate analysis showed that diabetes was associated with worse function two years postoperatively [58]. In contrast, one univariate [66] and three multivariate analyses found that diabetes did not influence postoperative function at one year [28], at three years [34], at 4.79 years [66] and at five years [41].

##### Gait Activities/Limitations

*Relative change in outcome*: One multivariate analysis found that diabetes was not predictive of improvement in walking distance one year [64] postoperatively.

*Absolute outcome*: Another multivariate analysis also found that diabetes was not predictive of walking limitations at six months [52].

##### Satisfaction

*Absolute outcome*: One multivariate analysis found that diabetes was associated with lower odds of being satisfied one year postoperatively [64] and one univariate analysis also reported that diabetes was not correlated with satisfaction 4.79 years postoperatively [66].

##### Quality of Life

*Relative change in outcome*: One multivariate analysis [68] found no influence of diabetes on QoL (SF-36) at two years [68]. Another multivariate analysis study identified diabetes to be a significant predictor for greater improvement in SF-12 MCS at one year, but not for the SF-12 PCS [28].

*Absolute outcome*: One univariate analysis stated that diabetes did not influence QoL two years postoperatively [58].

##### Conclusion

Concerning the influence of diabetes on postoperative pain, functional impairments, functional activities and gait activities, studies reported conflicting results, however results were never in favour of patient with diabetes meaning that patients with diabetes did never show less pain, less functional impairments and limitations or better gait functions compared to patients without diabetes. Two studies describing the influence of diabetes on QoL showed conflicting results, either in favour of diabetes or similar results for both patients with and without diabetes. No studies reported results on gait impairments.

#### 3.4.3. Factor: Cytokine Levels

##### Outcome Measure: Pain

*Relative change in outcome*: One multivariate analysis demonstrated that higher synovial fluid concentrations of TNF-a, MMP-13 and IL-6 were independent predictors of less pain improvement two years postoperatively [36]. Another multivariate analysis found that miRNAs were no independent predictors of postoperative pain relief one year postoperatively [38].

*Absolute outcome*: More severe preoperative synovitis, which is associated with higher levels of proinflammatory cytokines, seems to be associated with less postoperative pain at one year according to the univariate analysis of Petersen et al. [51] and Sideris et al. [67]. Another multivariate analysis found IL-1B and TNF-a to be independent predictors of greater pain development 6 months postoperatively [57].

##### Conclusion

Five studies examined the influence of cytokines on postoperative pain and their results were inconsistent. No results were found for other outcome measures.

#### 3.4.4. Factor: Dyslipidaemia

##### Pain

*Absolute outcome*: Dyslipidaemia was not predictive of postoperative pain at six months according to one multivariate analysis study [52].

##### Functional Impairment

*Absolute outcome*: According to the multivariate analysis of Pua et al. dyslipidaemia was not predictive of postoperative knee ROM at six months [52].

##### Functional Activities/Limitations

*Absolute outcome*: Hypercholesterolemia seems not predictive for a diminished functional outcome at one year [35].

##### Gait Activities/Limitations

*Absolute outcome*: Pua and colleagues found that dyslipidaemia was not predictive for postoperative walking limitations six months postoperatively [52].

##### Conclusion

The presence of dyslipidaemia appears to have no influence on postoperative TKA outcome according to two multivariate analyses [35,52]. 

## 4. Discussion

The present study systematically reviewed the scientific literature regarding the influence of metabolic and inflammatory factors on pain, function (impairments, activities, gait impairments and gait activities), satisfaction and QoL after TKA. Conflicting results (level 3 of conclusion) were found for the role of BMI, diabetes, cytokine levels and dyslipidaemia on postoperative TKA outcome. Possible explanations for these conflicting results will be discussed below.

### 4.1. Body Mass Index

Conflicting results were found for the role of BMI as an influencing factor on pain, function, satisfaction and QoL. These conflicting results can find their origin in several factors. First, BMI was not measured in a uniform manner. Some studies gathered BMI by self-report, others extracted BMI from patient records and only few studies measured weight and height of the patients themselves. This could possibly result in inaccurate BMI classification of patients. Second, the use of different classification methods for obesity could also have influenced the results. For example, De Leeuw et al. defined patients as non-obese if they had a BMI lower than 25 kg/m^2^ [31], while Dettoni et al. used a cut-off of 30 kg/m^2^ [33].

### 4.2. Diabetes

A possible explanation for the inconsistent results is the diversity in defining diabetes. Some studies relied on self-report, others used medical records, but diabetes was never defined by blood results which could have led to misclassifications. Further, all included studies defined diabetes as a dichotomous variable, and no studies using continuous variables of glycaemic control, for example HbA1C, were found. Next, the functional impact of diabetes on a patient’s life, the duration of diabetes and the presence of diabetic complications (such as neuropathy, nephropathy, etcetera) could also have been useful to consider as influencing factors on TKA outcome [69].

### 4.3. Cytokine Levels

Two multivariate analyses found greater peripheral blood concentrations of IL-1B and TNF-a [57] and greater synovial fluid concentrations of TNF-a, MMP-13 and IL-6 [36] to be independent predictors of postoperative pain development. This can be explained by the fact that pro-inflammatory cytokines can sensitize the peripheral nerve endings leading to preoperative peripheral and central sensitization [14], which has found to be associated with postoperative pain after TKA [70]. In contrast, the univariate analyses of Petersen et al. and Sideris et al. [67] found that more severe preoperative synovitis was associated with less postoperative pain [51]. However, these contrasting results can be explained by the fact that both studies did not control for other possible influencing preoperative factors.

There was also one study exploring the role of preoperative microRNAs on postoperative outcome [38]. These microRNAs are directly involved in the production of cytokines and are therefore included in this review [71,72]. The study of Giordano et al. found higher levels of certain microRNAs (hsa-miR-146a-5p, hsa-miR-145-5p and hsa-miR-130b-3p) to be associated with lower postoperative pain relief [38]. This was found using t-tests, but when performing linear regression analyses and including preoperative pain intensity in the model, a known risk factor for poor outcome, only a trend to significance (*p* = 0.06) of hsa-miR-146a-5p was found. Clearly, this points towards the importance of taking into account the already known risk factors. Interestingly, it appears that high levels of TNF-a and IL-1B induce the expression of hsa-miR-146a-5p and this pathway is involved in the pathogenesis of OA [73,74]. Since Giordano et al. did not include these pro-inflammatory cytokines in their regression analyses together with microRNAs [38], it is difficult to assess whether pro-inflammatory cytokines or microRNAs are predictive of poorer postoperative outcome.

### 4.4. Strengths and Limitations

This study had several strengths. First, a comprehensive set of search terms was used to search three databases for relevant studies. Second, the screening was performed by two independent reviewers. Besides the strengths of the current systematic review, there are also some limitations It was difficult to compare the results, due to heterogeneity in the different studies. Many different outcome measures at different follow up times were used, which might (partly) explain the, sometimes conflicting results. The risk of bias was not scored in a double blinded way. Finally, very limited studies were found concerning the influence of cytokine levels and dyslipidaemia.

### 4.5. Implications for Further Research and Clinical Practice

A better understanding of specifically the role of BMI, diabetes, inflammation and dyslipidaemia in postsurgical chronic pain, function, QoL and patient satisfaction after TKA is crucial to gain more insights into the timing of surgery, (p)rehabilitation, patient expectations, and patient-surgeon shared decisions. Therefore large-scaled longitudinal studies assessing the predictive value of metabolic and inflammatory factors pre-surgery in addition to the already evidenced risk factors with follow-up of 1 year after TKA are warranted. Retrospective studies could guide future researchers in the selection of these metabolic and inflammatory factors.

This insight could help us to identify those patients most at risk for chronic postoperative pain and disability, so that treatment strategy can be adapted and optimized, and outcome after TKR will be better, e.g., by providing (p)rehabilitation strategies specifically targeting these metabolic and inflammatory factors.

## 5. Conclusions

Reporting of study findings was challenging, because of the heterogeneity of the included studies. In conclusion, studies reported conflicting results regarding the influence of BMI on postoperative outcome in favour of obese as well as non-obese patients. The influence of diabetes on TKA outcome was also unclear, however results were never in favour of patient with diabetes. There were inconclusive results regarding the influence of cytokines. And finally, the presence of dyslipidaemia appears to have no influence on postoperative TKA outcome. Further research including larger patient cohorts unravelling the predictive role of BMI, diabetes, inflammation and dyslipidaemia in addition to the already known risk factors for poor outcome after TKA is required to identify a more comprehensive insight in possible risk factors and to provide the best possible care for patients with end-stage knee OA, undergoing TKA.

## Figures and Tables

**Figure 1 ijerph-20-05796-f001:**
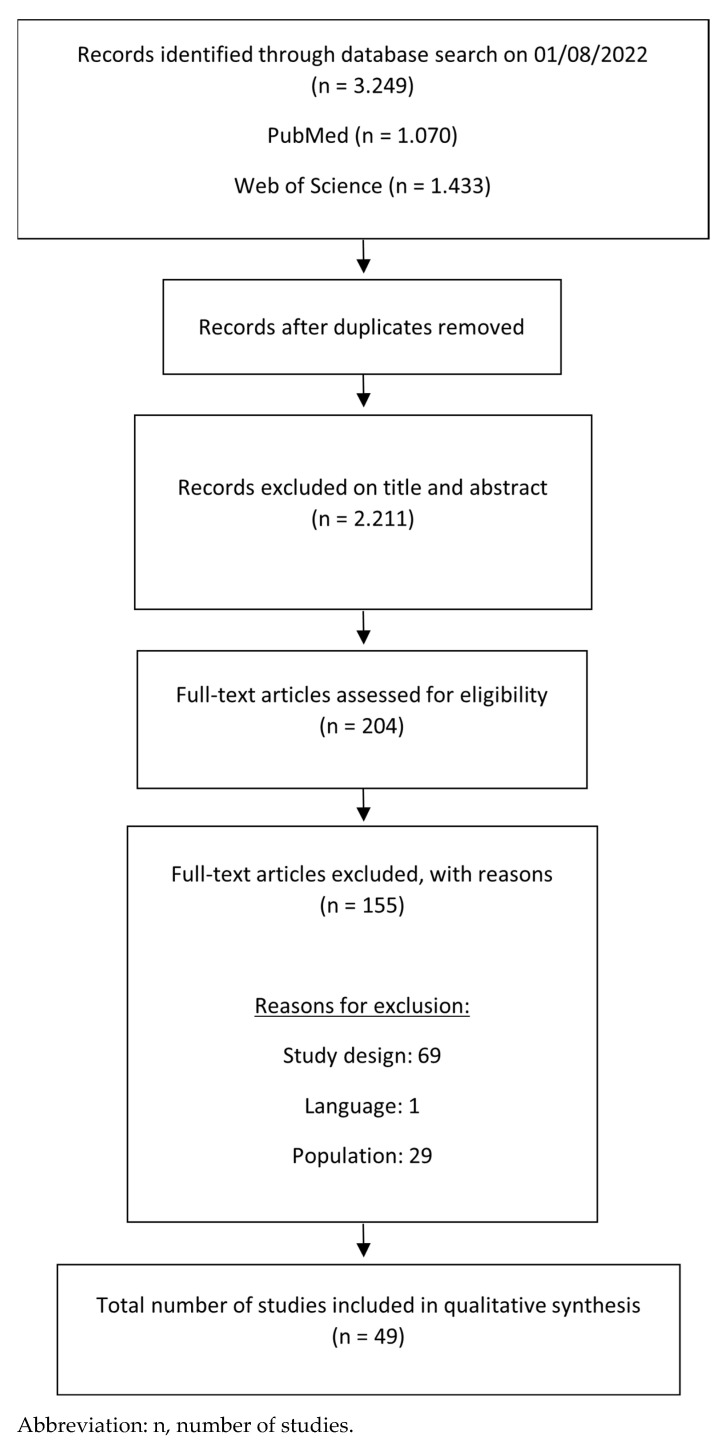
Flow chart of systematic review.

**Table 1 ijerph-20-05796-t001:** Inclusion and exclusion criteria.

Inclusion		Exclusion
Humans	Population	Animals
Knee OA patients scheduled for a primary TKA		Patients with RA
		Patients undergoing partial knee replacement or revision surgery
Preoperatively measured metabolic factors or inflammatory markers such as: obesity, diabetes, comorbidities, hs-CRP, cytokines	Predictive factor	Other than metabolic factors or inflammatory markers
Pain, functional ability, satisfaction or QoL measured ≥6 months postoperative	Outcome	Pain, functional ability, satisfaction or QoL measured <6 months postoperative Outcomes not related to pain, functional ability, satisfaction or QoL
Full text reports of original research (RCTs, prospective cohort studies, case-control studies)	Study design	Other designs (e.g., reviews, letter to the editor, and editorial papers, single case reports, retrospective studies) No full text available
English, Dutch, French	Language	Other languages

OA = Osteoarthritis; TKA = Total Knee Arthroplasty; RA = Rheumatoïd Arthritis; hs-CRP = high sensitive C-reactive Proteïne; QoL = Quality of Life.

**Table 2 ijerph-20-05796-t002:** Characteristics of the included studies.

Source, Study Design & Origin	Participants			Predictive Factors (BMI Value Expressed in kg/ m^2^)	Postoperative Follow-Up Time (≥6 m)	Outcome Measures	Analysis Univariate/Multivariate (Confounding Factors Are Given If Mentioned)	Results (+Effect Size If Result Is Statistically Significant and If Mentioned)
	Sample Size (nr. of TKAs) Mean Age (SD or Range) Sex	Inclusion	Exclusion					
Amusat et al. (2014) [24] Prospective, observational study Canada	N = 405 68y (10) 249 ♀ (62%) 156 ♂ (38%)	>40y Primary TKA Residing within the health region English speaking	Hemiarthroplasties Unicompartmental revisions Emergency arthroplasties	Diabetes (type I or II not specified) A. No diabetes B. Diabetes without functional impact C. Diabetes with functional impact	6 m	WOMAC pain WOMAC function	Multivariate (diabetes status, baseline WOMAC pain and function, depression, kidney disease, MOS social support score, HUI3 score, other weight-bearing joint involvement, age, gender)	Group C is associated with less pain reduction (β = 8.28, 95% CI (4.05 to 12.51), *p* < 0.001) Group C is associated with less functional improvement (β = 5.42, 95% CI (1.39 to 9.46), *p* < 0.01)
Ayers et al. (2022) [63] Prospective, cohort study USA	N = 4402 Satisfied group (N = 3843) 67.2 y (8.6) 1421 ♀ (37%) 2422 ♂ (63%) Dissatisfied group (N = 559) 66 y (8.9) 190 ♀ (34%) 369 ♂ (66%)	Primary unilateral TKA	Revision or bilateral procedures	BMI	5 y	Satisfaction (5-point Likert scale)	Multivariate (age, gender, race, BMI, CCI, marital status, smoking status, education level, insurance coverage, number of other painful hip and knee joints, and PROMs including preop ODI, KOOS pain and ADL scores, and SF-36 MCS and PCS scores)	BMI was no independent predictive factor for patient dissatisfaction
Bin Abd Razak et al. (2016) [25] Prospetive, observational study Singapore	N = 3062 66.4y (8.0) 2434 ♀ (79.5%) 628 ♂ (20.5%)	Primary unilateral TKA Other knee being asymptomatic or successfully replaced Completed all appropriate FU appointments and outcome assessments	Spastic or flaccid paralysis of one or both lower limbs regardless of cause New York Heart Association Class II and III cardiac failure Severe pulmonary disorders limiting the patient to only home ambulation All revision arthroplasties including infected arthroplasties Severe hip and/or spine conditions preventing patient from walking independently	BMI	5 y	A “good outcome” = an overall improvement in the outcome scores greater than or equal to the MCID. The calculated MCID for this cohort of patients was 5 for the OKS and 10 for the PCS	Multivariate (age, BMI, pre-op flexion range, KSS, OKS, MCS score, PCS score, mechanical alignment)	BMI is no predictors for a “good outcome” (*p* > 0.05)
Bonnefoy-Mazure et al. (2017) [26] Prospective, observational study Switzerland	N = 79 A. BMI < 30 kg/m^2^: N = 45 69.5 y (6.9) 27 ♀ (60%) 18 ♂ (40%) B. BMI ≥ 30 kg/m^2^: N = 34 67.0 y (7.8) 24 ♀ (70.5%) 10 ♂ (29.5%)	Unilateral TKA for symptomatic end-stage knee OA KL grade III-IV	Previous knee, hip, or ankle arthroplasty surgery History of lower limb or back surgery Neurologic or orthopedic disorders that could affect gait or balance If they used crutches or similar walking aids for short distance walking	BMI: A. Non-obese: BMI < 30 B. Obese: BMI ≥ 30	1y	Gait outcomes: Gait velocity Knee flexion ROM during gait cycle Clinical outcomes: WOMAC pain WOMAC function SF-12 MCS SF-12 PCS	Univariate: clinical outcomes Multivariate: gait outcomes (age, gender, WOMAC pain improvement)	Univariate: Similar gain non-obese vs. obese for: - Gait velocity (*p* = 0.353) - Knee flexion ROM during gait cycle (*p* = 0.860) - WOMAC function (*p* = 0.055) - SF-12 MCS (*p* = 0.717) - SF-12 PCS (*p* = 0.481) Larger gain obese group vs. non-obese for WOMAC pain: (r = 0.59, *p* = 0.011) Multivariate: No association between BMI and gait velocity gain (*p* = 0.353) or gain of knee ROM during gait cycle (*p* = 0.861)
Çankaya et al. (2016) [27] Prospective, observational study Turkey	N = 70 67.3 y (8.0) 53 ♀ (76%) 17 ♂ (24%)	Unilateral primary knee OA	Rheumatological joint diseases Previous knee surgery Metabolic bone disease OA in the contralateral knee	BMI: A. BMI < 30 B. BMI ≥ 30	6 m	KOOS-PS SF-36 PCS	Univariate	No relationship between BMI (*p* = 0.098) and KOOS-PS change. Lower BMI → greater increase in SF-36 PCS (*p* = 0.041)
Christensen et al. (2020) [61] Prospective, observational study (secondary analysis) USA	N = 65 57.7 y (5.9) 33 ♀ (51%) 32 ♂ (49%)	1. ≤65 y Diagnosed with end-stage knee OA Underwent an uncomplicated primary TKA by 1 of 3 fellowship-trained orthopedic surgeons	No criteria	BMI	1 y (13.2 m ± 0.3 m)	Physical function Satisfaction (5-point Likert scale)	Multivariate (BMI, sex, physical activity level, number of comorbidities, depression, expectations, pain interference)	BMI was not related to post-op physical function or satisfaction
Clement et al. (2013) [28] Prospective, observational study Scotland	N = 2389 Diabetes patients: 70.4 y (9.5) Non diabetes patients: 70.1 y (8.5) 1375 ♀ (57.5%) 1014 ♂ (42.5%)	No info	No info	Diabetes (type I and II)	1 y	OKS SF-12 PCS MCS	Multivariate (comorbidity, pre-op OKS, pre-op SF-12)	No influence of diabetes on OKS (*p* = 0.54) and SF-12 PCS (*p* = 0.22) but diabetes patients had a larger improvement in SF-12 MCS compared to patients without diabetes (β = 1.29, 95% CI (0.10 to 2.48), *p* = 0.03)
Collins et al. (2017) [29] Prospective, observational study United States	N = 633 65.9 y (8.5) 375 ♀ (59.2%) 258 ♂ (40.8%)	English-speaking ≥40 y TKA Primary diagnosis of OA	Diagnoses other than OA (e.g., inflammatory arthritis) Dementia Unicompartmental knee arthroplasty Bilateral TKA	BMI A. Normal weight: <25 B. Overweight: 25–29.9 C. Class I obese (moderate): 30–34.9 D. Class II obese (moderate): 35–39.9 E. Class III obese (gross/morbid) ≥ 40	6 m 24 m (=2 y)	WOMAC pain WOMAC function Patient satisfaction (“how satisfied are you with your operated knee?”)	Multivariate (age, sex, race, diabetes, musculoskeletal functional limitations index, pain medication use, study site)	During 6–24 m interval: all BMI groups experienced similar improvement in WOMAC pain (*p* = 0.5936) and WOMAC function (*p* = 0.5525) At 24m: no differences in WOMAC pain (*p* = 0.2996), WOMAC function (*p* = 0.2153) and satisfaction (*p* = 0.8246) across all BMI groups
Collins et al. (2012) [30] Prospective, observational study United Kingdom	N = 385 Non-obese (BMI < 30): 66.4 y 55 ♀, 132 ♂ Mildly obese (BMI 30–35): 66.6 y 39 ♀, 68 ♂ Highly obese (BMI ≥ 35): 62 y 39 ♀, 12 ♂	Primary TKA	No criteria	BMI: A. Non-obese: <30 B. Mildly obese: 30–34.9 C. Highly obese: ≥35 D. Obese: B + C	T1 = 6 m T2 = 18 m T3 = 3 y T4 = 5 y T5 = 9 y	KSS: - knee - function	Univariate	Within group differences at T5 (9 y) KSS knee and function were higher than the pre-op scores (*p* < 0.001) in group A, B and C except for KSS function in C (*p* = 0.053). Between group differences at T5 (9 y) KSS knee: group B and D had worse knee scores at than group A (*p* = 0.023 and *p* = 0.008 respectively). Group C did not (*p* = 0.086) KSS function: group B,C and D had worse function scores than group A (*p* = 0.007, *p* = 0.001 and *p* < 0.001 respectively) Outcome at earlier FU (T1–T4): Group D had worse KSS knee and function scores than group A at T1-T4 (*p* < 0.05) Group B had worse knee scores than group A at T1 (*p* = 0.034), T4 (*p* = 0.023) Group C only had worse KSS knee scores at T1 (*p* = 0.048) and T4 (*p* = 0.008), and lower KSS function scores at T1–T4 (*p* < 0.05)
Cooper et al. (2017) [22] Case-control study United States	N = 317 62.3 (9.5) 173 ♀ (54.4%) 144 ♂ (45.6%)	TKA for primary knee OA	Patients with a chronic pain condition other than knee OA Patients who were not naïve to TENS Patients with a condition that limited their participation ability to participate such as having a history of stroke or being wheelchair-bound	BMI	6 m	KOOS ADL Gait speed Daily activity: steps/day	Multivariate (age, sex, BMI, depression, state and trait anxiety, pain catastrophizing, knee flexion ROM category, pain with knee flexion, pain with knee extension, pain with gait speed testing, PPT, SF-36 PCS scores)	KOOS ADL: lower BMI pre-op predicted better perceived function at 6m FU (*p* = 0.005) Gait speed: lower BMI pre-op predicted higher gait speed at 6 m FU (*p* < 0.001) Steps/day: lower BMI pre-op predicted greater daily step count at 6 m FU (*p* = 0.001)
De Leeuw et al. (1998) [31] Prospective, observational study United Kingdom	N = 90 56 ♀ (62.2%) 34 ♂ (37.8%) Non-obese (BMI < 25): 71.9 y (59–84) Mildly obese (BMI 25–29.9): 71 y (51–80) Moderate and gross obese (BMI > 30): 67.3 y (38–80)	Primary TKA for knee OA	No info	BMI: A. Non-obese: BMI < 25 B. Grade I obese (mild): BMI 25–29.9 C. Grade II obese (moderate): BMI 30–39.9 + Grade III obese (gross/morbid) BMI > 40	1 y	Rosser Index Matrix (QoL)	Univariate	QoL improved in all groups (*p* < 0.01) with group B and C showing superiority in improvement (*p* < 0.01)
Deshmukh et al. (2002) [32] Prospective, observational study United Kingdom	N = 180 68.8 y (40–89) 95 ♀ (52.8%) 85 ♂ (47.2%)	Primary TKA for OA	Patients with TKA on the other side	BMI	1 y	NHP KSS - function - knee	Multivariate (age, sex, side of arthritis, medical comorbidity, pre-op NHP and KSS scores)	BMI did not influence the TKA outcome No further specific info at 1 y post-op was given
Dettoni et al. (2018) [33] Prospective, observational study Canada	N = 334 Age: no info Sex: no info	TKA for primary knee OA	Post-traumatic arthritis Severe post-op complications (infections, aseptic loosening)	BMI: A. Normal weight BMI < 30 B. Overweight/obese BMI 30–35 C. Highly/morbidly obese BMI > 35	6 m 1 y 2 y	KSS - knee - function - KSS total WOMAC	Multivariate (baseline values, BMI, tibial component measure)	No difference was reported at all endpoints between the 3 BMI groups (*p* > 0.05)
Gandhi et al. (2010) [34] Prospective, observational study Canada	N = 551 67.4 y (9.8) 349 ♀ (63.4%) 202 ♂ (36.6%)	>18 y Primary knee OA Minimum 1 y FU	No criteria	BMI	3 y (1–8 y)	WOMAC SF-36: - SF-36 role physical (RP) score - SF-36 physical function (PS)	Multivariate (age, gender, ethnicity, BMI, comorbidity, level of education)	BMI was not predictive of a less sustained functional outcome on the WOMAC scale(*p* = 0.64) or the SF-36 (*p* = 0.95)
Gandhi et al. (2010) [35] Prospective, observational study Canada	N = 889 Age: No info Sex: No info	>18 y Primary or secondary OA Unilateral TKA	Patients having 0 metabolic abnormalities	BMI Diabetes (type II) Hypertension Hypercholesterolemia MetS (=BMI > 30 kg/m^2^ + patient self-reported diagnosis of hypercholesterolemia, hypertension, and diabetes)	1 y	WOMAC	Multivariate (age, sex, baseline total WOMAC scores, comorbidity)	The number of MetS risk factors was not predictive of total WOMAC scores (*p* > 0.05) For the models where the individual metabolic factors were entered, only obesity predicted diminished outcome (β = 3.6, 95% CI (0.02 to 7.2), *p* = 0.04)
Gandhi et al. (2013) [36] Prospective, observational study Canada	N = 28 68.5 y (9.4) 16 ♀ (57%) 12 ♂ (43%)	End stage OA undergoing TKA Diagnosis of OA based on the ACR criteria: knee pain and radiographic osteophytes and at least one of the following 3: age > 50 years, morning stiffness ≤30 min in duration, or crepitus	No criteria	Serum and synovial fluid cytokine levels: IL-6 IL-1β MMP-9 MMP-13 MIP-1β MCP-1 Adiponectin Leptin TNF-α IFN-γ VCAM-1	2 y	WOMAC pain	Multivariate (age, gender, BMI, comorbidity count)	Greater synovial fluid concentrations of TNF-a (*p* = 0.001), MMP-13 (*p* = 0.03) and IL-6 (*p* = 0.001) were independent predictors of less pain improvement
Giesinger et al. (2018) [37] Prospective, observational study United Kingdom	N = 402 70.7 y (9.2) 222 ♀ (55.2%) 180 ♂ (44.8%)	Primary TKA	No criteria	BMI A. Normal weight: BMI < 25.0 B. Overweight: BMI 25.0–29.9 C. Class I obesity: BMI 30.0–34.9 D. Class II obesity: BMI 35.0–39.9. E. Class III obesity: BMI ≥ 40.0	1y	OKS EQ-5D-3L Treatment satisfaction: (very satisfied, satisfied, unsure, dissatisfied, very dissatisfied)	Univariate	OKS scores are associated with BMI at 1 y (*p* < 0.001) with group A and B obtaining highest scores, while group E showed lowest scores Improvement of OKS scores from pre-op to 1y post-op did not differ across BMI groups EQ-5D-3L: lower BMI scores were associated with better general health (*p* < 0.001) Improvement of general health did not differ across BMI groups Treatment satisfaction: BMI groups differed in post-op treatment satisfaction (*p* = 0.029) in favour of the less obese groups
Giordano et al. (2020) [38] Prospective, observational study Denmark	N = 136 High pain relief group: 69.03 y (8.7) Low pain relief group: 68.00 y (10.1) 82 ♀ (60.3%) 54 ♂ (39.7%)	Knee OA Scheduled for TKA	Patients with other diagnosed pain conditions (e.g., hip OA, rheumatoid arthritis, fibromyalgia, and neuropathic pain), sensory dysfunction, or mental impairment	miRNAs	1 y	Pain (VAS)	Multivariate (pre-op pain intensity, miRNAs)	There were no microRNAs found to be an independent predictor of post-op pain relief (*p* > 0.05)
Hakim et al. (2020) [39] Prospective, observational study Israel	N = 374 64.3 y (48–83 y) ♀ no info ♂ no info	Patients with well-balanced hypertension or diabetes mellitus and other medical conditions Perceived primary TKA for primary knee OA Patients with BMI > 30 kg/m^2^ on the day of surgery Patients with minimum FU of 4 years	Patients with post-traumatic knee OA, including previous fractures or dislocation, knee instability, and post-menisectomy patients with a history of various rheumatic diseases patients with incomplete clinical or radiographic records	BMI: A. Non-obese <29.9 B. Obese: 30.0–39.9 C. Morbid obese: >40.0	10.8 y (4–17 y)	KSS KSS function	Univariate	KSS - all groups → post-op improvement with higher improvement in A vs. B and C (F = 8.89, *p* < 0.001) - post-op KSS differed between A and C (*p* = 0.046) and between B and C (*p* = 0.030) in favour of the less obese patients. - post-op KSS did not differ between A and B (*p* = 0.530) KSS function: - all groups →post-op improvement - post-op KSS function differed between A and C (*p* = 0.011) and between B and C (*p* = 0.001) in favour of the less obese patients. - post-op KSS function did not differ between A and B (*p* = 0.700)
Hodges et al. (2018) [62] Prospective, observational study (secondary analysis) Australia	N = 349 65 y (6.3) 185 ♀ (53%) 164 ♂ (47%)	Patients between 45 y and 74 y Scheduled to undergo unilateral or bilateral TKA	Patients with previous (last 12 months) or anticipated (next 6 months) joint replacement surgery Major comorbidity preventing aerobic exercise at 50–60% of maximum heart rate Rheumatoid arthritis Major neurological conditions	BMI	1 y	Physical activity: Active Australia survey Sedentary behavior: “how many hours in 24 h do you spend sitting?”	Multivariate (age, sex, usual care, obesity, knee pain, activity limitations, knee extensor strength, comorbidity score, psychological well-being, lack of sleep, lack of energy, fatigue)	Obesity (β = 1.54, 95% CI (0.96 to 2.48), *p* = 0.07) was an independent predictor of inadequate physical activity at 1 y BMI was not predictive for sedentary behavior (*p* > 0.1)
Järvenpää et al. (2012) [40] Prospective, observational study Finland	N = 52 76.3 y (6.7) 40 ♀ (83.3%) 8 ♂ (16.7%)	No previous knee or hip operations No medication or diseases known to influence bone mineral metabolism	Patients participating in another TKA study	BMI A. Non-obese: BMI < 30 B. Obese: BMI ≥ 30	1 y 10.8 y (9–12 y)	KSS (1 y and 10.8 y): - Knee - function ROM (1 y and 10.8 y) WOMAC (10.8 y) Walking distance (10.8 y) TUG (10.8 y)	Univariate	Results at 1 y: - KSS knee did not differ between group A and B. - KSS function was lower in group B (*p* = 0.019) - ROM was lower in group B (*p* = 0.029) Results at 10.8 y: - Group B had lower KSS knee (*p* = 0.010), KSS function (*p* = 0.019), ROM (*p* = 0.016) and WOMAC-scores (WOMAC pain (*p* = 0.021); WOMAC stiffness (*p* = 0.006); WOMAC (*p* = 0.003)) - KSS function, walking distance and TUG was similar between group A and B (*p* > 0.05)
Jauregui et al. (2016) [41] Prospective, observational study United States	N = 287 knees 66 y 173 ♀ (61.6%) 108 ♂ (38.4%)	No criteria	Presence of a known neuromuscular or neurosensory deficit <18 y BMI > 40 kg/m^2^	BMI Diabetes (type I or II not specified)	5 y	- KSS functional (KSS F) - KSS objective (KSS O) - KSS combined (KSS C) (=functional + objective)	Multivariate (age, BMI, gender, race, alcohol consumption, level of education, school degree, tobacco use, comorbidities)	BMI Higher BMI → negative impact on KSS F and C (*p* < 0.001), not on KSS O (*p* = 0.068) Diabetes No association between diabetes and outcome
King et al. (2021) [64] Prospective cohort study (secondary analysis) Canada	N = 1051 67 y (9) 617 ♀ (58.7%) 434 ♂ (41.3%)	≥30 y Primary knee OA Primary TKA Read and comprehend English Attended 1y FU visit	Inflammatory arthritis	Diabetes (type I or II not specified)	1 y	Pain (WOMAC pain subscale) Function (KOOS) Patient Acceptable Symptom State (PASS) Substudy (N = 278): 6MWT	Multivariate (age, sex, smoking status, BMI, education, social support)	Pain and Function: diabetes was not associated with reduced improvement of pain/ function. Patient Acceptable Symptom State (PASS): diabetes was associated with lower odds of reporting acceptable symptom state (OR 0.64, 95% CI 0.44–0.94) 6MWT: diabetes was not associated with less improvement in walking distance
Lamb et al. (2003) [42] Prospective, observational study United Kingdom	N = 58 71.1 y (6.4) 27 ♀ (47%) 31 ♂ (53%)	Primary unilateral knee OA Patients on the waiting list for surgery	No criteria	BMI	6 m	Walking speed Stair climbing speed	Multivariate (age, gender, comorbidities, pain, BMI, total leg extensor power in both legs, flexion)	Walking speed was not predicted by pre-op BMI (*p* = 0.06) Stair climbing speed was predicted by pre-op BMI (*p* = 0.017)
Lampe et al. (2016) [20] Prospective RCT (secondary analysis) Germany	N = 100 69.1 y (7.8) 73 ♀ (73%) 27 ♂ (27%)	Clinical and radiological signs of knee OA with failed non-operative treatment No indication for a uni-compartmental implant or joint-preserving osteotomies Age from 40–90 y ASA pre-op classification grade 1–3 No deformity larger than 20° varus or 15° valgus No previous bone surgery to the index knee No previous total joint replacement at the index leg No post-op infection of the index knee or thrombosis within the FU period	No criteria	BMI	4 y	KSS function	Multivariate (different surgically modifiable factors and patients-specific factors)	The combination of BMI, age, pre-op KSS-F, tibial component slope and femoral offset changes medial predicted the 4 y KSS-F (*p* = 0.007) Lower BMI in this model led to better KSS-F
Lampe et al. (2016) [21] Prospective RCT (secondary analysis) Germany	N = 100 69.1 y (7.8) 73 ♀ (73%) 27 ♂ (27%)	Clinical and radiological signs of knee OA with failed non-operative treatment No indication for a uni-compartmental implant or joint-preserving osteotomies Age from 40–90 y ASA pre-op classification grade 1–3 No deformity larger than 20° varus or 15° valgus No previous bone surgery to the index knee No previous total joint replacement at the index leg No post-op infection of the index knee or thrombosis within the FU period	No criteria	BMI	1 y	Maximal knee flexion	Multivariate (different surgically modifiable factors and patients-specific factors)	The combination of pre-op maximal knee flexion and BMI predicted the 1 y maximal knee flexion (*p* < 0.001) Lower BMI in this model led to better maximal knee flexion
Li et al. (2017) [43] Prospective, observational study United States	N = 2964 69 y ♀ (61.1%) ♂ (38.9%)	Primary diagnosis of knee OA Both pre-op and 6m post-TKA functional outcome data and valid body weight and height data at the time of the surgery were available	Another diagnosis then knee OA (for example, osteonecrosis or inflammatory arthritis) TKA for an acute fracture or cancer	BMI A. Under or of normal weight (<24.99) B. Overweight (25.00 to 29.99) C. Obese (30.00 to 34.99) D. Severely obese (35.00 to 39.99) E. Morbidly obese (≥40.00)	6m	SF-36 PCS KOOS pain	Multivariate (baseline function and pain score, sex, age, race, household income, education, living alone, type of insurance, medical comorbidities, low back pain, number of other painful joints, surgical volume of the hospital)	SF-36 PCS: - Greater level of obesity → worse PCS scores at 6 m (*p* < 0.001) - Similar change in the PCS score between baseline and 6m for all BMI groups KOOS pain: - At 6 m, the pain scores were excellent regardless of BMI status and the mean pain scores were in a very close range, except for group E, whose mean score was slightly lower (worse) than the scores in the other groups (*p* = 0.02). - Greater level of obesity → larger improvements between baseline and the 6m post-TKA pain scores (*p* < 0.001)
Lizaur-Utrilla et al. (2014) [44] Prospective matched study Spain	Obese N = 171 70.2 y (43–81) 111 ♀ (76) 60 ♂ (24) Non-obese N = 171 70.7 y (45–83) 111 ♀ (76%) 60 ♂ (24%)	OA diagnosis Obese group: BMI of ≥30 Control group: BMI < 30 Pre-op knee function KSS (±5 points)	Diagnosis of inflammatory arthritis	BMI A. Non-obese: <24.9 B. Class 0: 25.0–29.9 (overweight) C. Class I: 30.0–34.9 D. Class II: 35.0–39.9 E. Class III: ≥40 (morbid)	5 y	KSS - Knee - Function WOMAC - Pain - function SF-12: - PCS - MCS	Univariate Multivariate (only for KSS outcome and included factors not further specified)	Univariate - KSS knee, WOMAC pain, and SF12 PCS and MCS → similar for obese and non-obese patients - KSS function (*p* = 0.013) and WOMAC function (*p* = 0.019) were better for non-obese patients - No differences between obese class I-II and class-III for all outcomes (*p* > 0.05) - Multivariate - No influence of the BMI (*p* = 0.166) on KSS outcome score
McQueen et al. (2007) [45] Prospective, observational study United States	N = 50 68 y (54–80) 36 ♀ (72%) 14 ♂ (38%)	TKA for primary knee OA	No criteria	BMI: A. Ideal body weight (<25 kg/m^2^) B. Overweight (25 to 30 kg/m^2^) C. Obese (>30 kg/m^2^)	6 m	WOMAC: - Pain - Stiffness - Physical functioning - Total score SF-36	Univariate	WOMAC pre-op vs. post-op: Group A: Similar WOMAC pain (*p* = 0.094) and stiffness (*p* = 0.229) but improvement of physical functioning (*p* = 0.021) and total score (*p* = 0.040) Group B: Similar WOMAC stiffness (*p* = 0.402) but improvement of pain (*p* = 0.004), physical functioning (*p* = 0.012) and total score (*p* = 0.008) Group C: Improvement of all WOMAC scores (*p* < 0.001) SF-36 pre-op vs. post-op: Group A: improvement in 1/10 SF-36 components: Role Limitation–Physical (*p* = 0.029) Group B: improvement in Physical Functioning (*p* = 0.001), Bodily Pain (*p* < 0.001 and PCS (*p* = 0.001) Group C: improvement in all components except General Mental Health (*p* = 0.053)
Merle-Vincent et al. (2011) [46] Prospective, observational study France	N = 264 75 y (7.8) 186 ♀ (70.5%) 78 ♂ (29.5%)	Knee OA meeting ACR criteria TKA scheduled on the following day Availability of a standard anteroposterior or schuss radiograph of the knees	OA of the target knee due to inflammatory joint disease, Paget disease, septic arthritis, or tuberculous arthritis Osteonecrosis of the target knee Pure chondrocalcinosis without evidence of OA Symptomatic hip OA on the same side as the target knee Inability or unwillingness to answer the study questions	BMI	2 y	Satisfaction rate (0/25/50/75/100%)	Multivariate (age, sex, BMI, radiological joint narrowing score, complications, feelings of depression at baseline and after 2 y, Lequesne index at baseline, change in Lequesne index after 2 y vs. baseline)	BMI < 27 kg/m^2^ predicts better satisfaction after 2 y (OR: 0.1, 95% CI (0.03–0.7), *p* = 0.015)
Mishra et al. (2022) [65] prospective study India	N = 100 No further info	Severe (Grade IV) OA or moderate (Grade III) OA with gross functional limitation Signed informed consent	Rheumatoid arthritis Patients who underwent previous uni-condylar knee replacement Patients who underwent Previous High Tibial Osteotomy (HTO). Haemophilic knee joint arthritis Gouty arthritis Patients having damage to the knee joint attributed to vascular aetiology Post-traumatic knee OA, including previous fractures or dislocation, knee instability, and post-menisectomy	BMI Normal weight: 18.5–24.9 Overweight: 25–29.9 Class I obese: 30–34.9 Class II obese: 35–39.9 Class III obese ≥40	6 m–1 y	VAS pain KSS FKSS PROMS (patients response outcome measures) Functional outcomes	Univariate	Pain: decreased with time in all the classes of obesity, with a maximum decrease in A and B and a minimum decrease in E (*p* < 0.001) KSS and FKSS: improved with time in all classes, with group E having a minor improvement (*p* < 0.001) PROMS: all classes of obesity had similar PROMS (*p* < 0.001). Improvement of PROMS was highest in Group E Functional outcomes: all classes of obesity had similar functional outcomes with no residual deformity (*p* < 0.001)
Nunez et al.(2011) [23] Case-control study Spain	Study group (class II and III obesity): N = 70.2 y (6.7) 7 ♀ (11.7%) 53 ♂ (88.3%) Control group: N = 71.7y (6.7) 7 ♀ (11.7%) 53 ♂ (88.3%)	Study group: Knee OA according to Kellgren and Lawrence criteria Severe and morbid obesity (BMI grades II ≥ 35 and III ≥ 40, respectively, according to the WHO classification Admitted to the knee unit for TKA between January 2006 and February 2007 Control group Each patient in the study group was matched according to age, sex, and total pre-op (baseline) WOMAC score with a patient with a BMI < 35 Admitted to the same knee unit for TKA	Functional illiteracy Psychopathology severe enough to impede total participation in study procedures	Control group: BMI < 35 Study group: A. Class II: BMI 35 to 39.9 B. Class III: BMI ≥ 40	1y	WOMAC - Pain - Stiffness - Function - Total	Univariate (ES calculated as mean change/SD of baseline results)	Post-op improvement for all WOMAC scores in the study group (*p* < 0.001) Pain: ES 1.9 Stiffness: ES 1.1 Function: ES 1.9 Total: ES 2.0 Post-op improvement for all WOMAC scores in the control group (*p* < 0.001) except for WOMAC stiffness (*p* = 0.071) Pain: ES 2.2 Function: ES 2.2 Total: ES 2.2 At 1y, there were no differences in WOMAC dimension scores between study and control group
Nunez et al. (2007) [47] Prospective, observational study Spain	N = 67 74.8 y (5.6) 54 ♀ (80.6%) 13 ♂ (19.4%)	Primary TKA with a diagnosis of knee OA grade IV (according to KL criteria)	Functional illiteracy Psychopathology severe enough to impede total participation in study procedures	BMI: A. Class I: 25.0 to 29.9 B. Class II: 30.0 to 34.9 C. Class III: 35.0–39.9	3y	WOMAC - Pain - Stiffness - Function	Multivariate (sociodemographic, clinical, intra-operative surgical, in-patient and post-op clinical variables)	Severe (Class III) obesity was associated with more pain (*p* = 0.049) but not with stiffness or function
Overgaard et al. (2019) [48] Prospective, observational study Sweden	N = 3327 69 y 1912 ♀ (58%) 1415 ♂ (42%)	TKA patients operated for knee OA	Patients who did not have both pre-op and 1y post-op patient reported outcome data and those who had died during the FU year	BMI: A. Normal weight: BMI < 25.0 B. Overweight: BMI 25.0–29.9 C. Class I obesity: BMI 30.0–34.9 D. Class II + III obesity: BMI ≥ 35.0	1y	KOOS - Pain - ADL	Multivariate 1. (age, sex) 2. (age, sex, ASA grade, pre-op KOOS pain and ADL function)	No effect of BMI on change in KOOS pain and ADL function when adjusting for age and sex No effect of BMI on change in KOOS pain (*p* = 0.7) but a statistically sign. effect (*p* = 0.004) on change in ADL function (2 points less improvement/10 higher BMI units)
Paterson et al. (2020) [49] Prospective o study Australia	N = 78 69.9 y (7.3) 39 ♀ (50%) 39 ♂ (50%)	Primary TKA Knee OA Patients on waiting list between March 2013 and March 2016	Patients unable to provide informed consent Unable to undertake gait analysis without gait aid	BMI A. <30 kg/m^2^ B. ≥30 kg/m^2^	2 y	Gait biomechanics - Peak knee frontal plane angle - Varus-valgus thrust excursion - Peak knee flexion angle in stance - Knee sagittal plane range of motion - Peak KAM - KAM impulse - Peak KFM	Multivariate (in-patient rehabilitation (yes/no) and baseline normalized walking speed)	Obesity did not influence changes in gait biomechanics
Paxton et al. (2016) [50] Prospective, observational study United States	N = 11084 68 y (IQR 62-75) 6861♀ (62%) 4223 ♂(38%)	Patients ≥ 18 y with OA who underwent primary unilateral TKA	Post-op complication (e.g., infection, deep vein thrombosis, or pulmonary embolisms) Revision within 3 y of the index procedure Termination of membership or death within 2 y of the index procedure If a patient had bilateral TKA procedures within 3 y of each other, neither procedure was included in the analysis as this could affect the patient’s physical activity level	BMI Diabetes (type I or II not specified)	1–2 y	Change in reported physical activity (minutes per week)	Multivariate (sex, age, BMI, race, diabetes status)	BMI: Increasing BMI levels were associated with lower change in physical activity (−5.9 min/week, 95% CI (−7.9 to −3.9), *p* = 0.003) Diabetes: not associated (*p* = 0.4)
Petersen et al. (2020) [51] Prospective observational study Denmark	N = 26 High pain group (VAS > 30 12 m post-op): N = 9 64 y (4) 5 ♀ (56%) 4 ♂ (44%) Low pain group (VAS ≤ 30 12 m post-op): N= 17 70 y (2) 9 ♀ (53%)8 ♂ (47%)	Symptomatic, primary knee OA according to the ACR criteria, radiographically confirmed	Other local (e.g., nerve root entrapment) or generalized pain conditions (e.g., fibromyalgia) Any sensory dysfunctions Other sign. musculoskeletal disorders (e.g., hip OA) Mental impairment Insufficient Danish language skills precluding an informed consent Contraindications for CE-MRI (CE-MRI was not performed if the patient had an estimated glomerular filtration rate < 60 mL/min/1.73m^2^)	Synovitis: - Contrast-enhanced MRI (CE-MRI) - Dynamic contrast-enhanced MRI (DCE-MRI) - Histologic	1 y	Pain: A. Low pain group: VAS ≤ 30 B. High pain group: VAS > 30	Univariate	More severe pre-op synovitis was associated with less post-op pain: - CE-synovitis (R = 0.455, *p* = 0.022) - Number of voxelsxME (R = −0.528, *p* = 0.007) - Number of voxelsxIRE (R = −0.511, *p* = 0.009) - histologic: trend towards significance (R = −0.384, *p* = 0.053)
Pua et al. (2019) [52] Prospective, observational study Singapore	N = 4026 68 y (7.5) 3003 ♀ (75%) 1023♂ (25%)	≥50 years old Unilateral TKA for knee OA	Patients who underwent revision knee surgery within 6 m post TKA Patients with a history of rheumatoid arthritis Patients with stroke or Parkinson’s disease	BMI Diabetes (type I or II not specified) Dyslipidemia	6 m	Knee ROM: - flexion - extension Knee pain Walking limitations	Multivariate (age, sex, contralateral knee pain, BMI, education level, ethnic group, hypertension, dyslipidemia, diabetes, caregiver available, pre-op walking aids, pre-op depression level, pre-op knee extension, pre-op knee flexion, pre-op knee pain, pre-op walking limitation, weeks from surgery to assessment, week 24 knee extension, week 24 knee flexion, week 24 knee pain, week 24 walking limitation)	ROM: - Absence of diabetes is predictive of better post-op knee extension (OR: 0.78, 95% CI (0.67, 0.90), *p* < 0.001) and knee flexion (OR: 0.72, 95% CI (0.63, 0.83), *p* < 0.001) - BMI and dyslipidemia were not predictive - Knee pain: - BMI, Dyslipidemia or Diabetes were not predictive - Walking limitations: - BMI, Dyslipidemia or Diabetes were not predictive
Rissolio et al. (2021) [66] prospective observational study USA	N = 648 72 y (7.4) 472 ♀ (72.5%) 176 ♂ (27.5%)	Primary TKA due to OA	Inflammatory of post-traumatic knee OA BMI > 45 kg/m^2^ Previous osteotomies Use of constrained condylar and rotating hinged implants All cases that cold not comprehensively be evaluated by analyzing the digital database of the hospital Last X-ray <12 m	Diabetes (type I or II not specified)	Mean FU: 4.79 y	WOMAC KOOS FJS-12 Satisfaction (Yes/No to a single direct question)	Univariate	No correlation between diabetes and WOMAC, KOOS, FJS-12 or satisfaction
Scott et al. (2016) [53] Prospective, observational study United Kingdom	N = 177 50 y (17–54) 99 ♀ (56%) 78 ♂ (44%)	Patients <55 y old	No criteria	BMI	1 y	Satisfaction	Multivariate (K&L scale, OKS, indication (OA with meniscectomy, OA multiply operated, OA other surgery, OA BMI > 40, post-traumatic OA, inflammatory arthropathy)	BMI >40 kg/m^2^ is not independently predictive for patient satisfaction (*p* = 0.424)
Sideris et al. (2022) [67] USA Prospective cohort study	persistant postop pain (PPP) group group: N = 15 65.6 y (4.1) 8 ♀ (53.3%) 7 ♂ (46.7%) minimal postop pain (MPP) group: N = 147 67.2 y (8.4) 83 ♀ (56.5%) 64 ♂ (43.4%)	Unilateral TKA for severe end-stage OA OA was deemed severe by radiologist or surgeon using descriptors of ‘severe narrowing’ and/or ‘bone on bone’	Contraindications to NSAIDs, acetaminophen, dexamethasone, or regional anesthesia History of daily opioid use of ≥6 weeks or any usage of non-prescribed Opioids Patients receiving a periarticular injection for postop pain A history/diagnosis of any rheumatic or autoimmune disease Post-traumatic OA Crystalline arthropathy American Society of Anesthesiologists physical status score > 3 Current pregnancy Any active infections or current antibiotic use	Cytokines measured in synovial fluid	6 m	Persistent pain: PPP: (NRS ≥ 4 at 6 m) MPP: minimal postop pain group (NRS ≤ 3 at 6 m)	Univariate	Patients in the MPP group showed higher pre-op IL-10 (pg/mL) compared to the PPP group (*p* = 0.04) MPP: median 0.2 (IQR 0.1–0.3) PPP: median 0.1 (IQR 0.1–0.2)
Steinhaus et al. (2019) [54] Prospective observational study USA	N = 2472 67.8 y 1472 ♀ (59.5%) 1000 ♂ (40.5%)	Knee OA was the primary diagnosis They underwent primary unilateral TKA Gave consent to participate in the registry Had completed pre-op and 2 y FU surveys	Secondary diagnoses Concomitant surgical procedures Patients who underwent reoperation, revision or contralateral surgery prior to 2 y FU Complications at 6 m adverse event survey Incomplete BMI data or EQ-5D responses	BMI A. Underweight (<18.50) B. Normal weight (18.50–24.99) C. Overweight (25.00–29.99 kg/m^2^) D. Obese class I (30.00–34.99) E. Obese class II (35.00–39.99) F. Obese class III (≥40.00)	2 y	EQ-5D-3L: - EQ-5D index (HRQoL) - EQ-VAS (overall health status)	Multivariate (age, sex, CDI, years of surgery, length of stay)	EQ-5D: class I (−0.02 ± 0.01) and class III (−0.05 ± 0.01) were associated with lower scores EQ-VAS: BMI was associated with negative effect estimates increasing in effect size from class I to class III obesity, with estimates of -2.31 ± 0.84, −3.27 ± 1.06 and −5.76 ± 0.75 resp. (*p* < 0.05)
Stevens-Lapsley et al. (2010) [55] Prospective, observational study United States	N = 140 65.3 y (9.2) 75 ♀ (54%) 65 ♂ (46%)	Unilateral TKA for primary OA	Uncontrolled hypertension Uncontrolled diabetes Symptomatic OA in the contralateral knee (defined as self-reported knee pain >4 on a 10-point verbal analog scale Other lower extremity orthopedic problems that limited function Neurologic impairment BMI > 40 kg/m^2^	BMI	6 m	Functional performance: - TUG - SCT - 6MWT Perceived functional ability: - SF-36 ◦ MCS ◦ PCS - KOS-ADLS	Multivariate (pre-op outcome measures, BMI)	KOS-ADLS was the only parameter influenced by BMI (sign F change = 0.012)
Sveikata et al. (2017) [56] Prospective, observational study Lithuania	N = 294 70.9 y (8.3) 243 ♀ (82.7%) 51 ♂ (17.3%)	Knee OA Primary TKA Patients speak native language Patients agreed to participate in the study	No criteria	BMI A. <30 kg/m^2^ B. 30–35 kg/m^2^ C. 35–40 kg/m^2^ D. ≥40 kg/m^2^	1y	WOMAC SF-12 - MCS - PCS	Multivariate (age, sex, BMI, level of education, social support)	BMI was not an independent predictor of outcome (*p* > 0.05)
Tchetina et al. (2020) [57] Prospective observational study Russia	N = 50 67.6 y (7.5) 37 ♀ (74%) 13 ♂ (26%)	Primary Knee OA Primary TKA	Decompensated chronic diseases Active infectious process and foci of chronic infection Neurocirculatory disorders lower extremities Opioid-type analgesic therapy prior to surgery	IL-B TNF-a	6 m	VAS	Multivariate	IL-1B (*p* = 0.011) and TNF-a (*p* = 0.01) were independent predictors of post-op pain development Higher expression → more pain
Teo et al. (2018) [58] Prospective, observational study Singapore	N = 905 65.9 y (7.7) 710 ♀ (78.6%) 195 ♂ (21.4%)	Unilateral TKA for KL grade 3-4 OA	Patients lost to FU Patients with secondary arthritis from posttraumatic inflammatory, and/or infective causes	Diabetes (type I or II not specified)	2y	ROM KSS OKS SF-36 - PCS - MCS	Univariate	Diabetes is associated with worse OKS (*p* = 0.002) and KSS function score (*p* = 0.001) but not with SF-36 or ROM
Torres-Claramunt et al. (2016) [59] Prospective, observational study Spain	N = 517 72 y (2) 526 ♀ (76.3%) 163 ♂ (23.7%)	Primary TKA Patients who had undergone primary TKA in both knees during the study period only participated with the data obtained from the first surgery	Patients with some kind of cognitive disorder or language barriers which might hinder the comprehension of the questionnaires	BMI A. <30 kg/m^2^ B. 30.0–35 kg/m^2^ C. ≥35.0 kg/m^2^	5 y	SF-36 KSS	Univariate	Absolute SF-36 score at 5 y: All SF-36 domains (with the exception of the general health domain) were sign. better in group A than group B/C and better in group B than group C (*p* < 0.05) Improvement SF-36: Similar improvement in the 3 groups (*p* > 0.05) Absolute KSS score at 5 y: All KSS domains were sign. better in group A than group B/C and better in group B than group C (*p* < 0.05) Improvement KSS: Similar improvement in the 3 groups (*p* > 0.05)
Zeni et al. (2010) [60] Prospective, observational study USA	N = 105 65.8 y (8.9) Sex: no info	End-stage knee OA in at least 2 compartments (KL score, ≥3) All of the TKAs were posterior cruciate ligament-sacrificing condylar implants with patellar resurfacing	Notable pain in the contralateral limb (maximum pain, ≥4 of 10 during daily activities Diagnosis of arthritis involving any other lower extremity joint Cardiovascular or neurological impairments, including peripheral neuropathies	BMI	2 y	Stair-Climbing Task (use of handrail and gait pattern)	Multivariate (age, BMI, quadriceps index, knee flexion ROM, KOS-ADLS score, time to complete stair climbing task, use of handrail)	BMI did not sign. contribute to the prediction of handrail use after TKA (*p* = 0.845)
Zhang et al. (2021) [68] Prospective, observational study Singapore	N = 2840 66.3 y (8.2) 2008 ♀ (70.7%) 832 ♂ (29.3%)	Primary TKA	Bilateral TKA TKA due to inflammatory arthritis Post-traumatic arthritis Malignancy Avascular necrosis	BMI Diabetes (type I or II not specified)	2 y	SF-36 PCS SF-36 MCS WOMAC KSS Knee KSS Function KSS ROM	Multivariate (age, gender, race, ischemic heart disease, stroke, cancer, respiratory disease, preoperative scores)	SF-36 PCS: - diabetes: no sign different improvement - BMI: higher BMI (*p* < 0.001) → greater SF-36 PCS improvement (coef: −0.14, CI 95% (−0.20 to −0.08) SF-36 MCS: - diabetes, BMI: no sign different improvement WOMAC: - diabetes, BMI: no sign different improvement KSS Knee: - diabetes: poorer improvement (*p* = 0.025, coef: −1.22, CI 95% (−2.28 to −0.15)—BMI: no sign different improvement KSS Function - diabetes,: no sign different improvement - BMI: higher BMI (p = 0.005) → greater Knee function improvement (coef: −0.21, CI 95% (−0.36 to −0.07) Knee ROM - diabetes: poorer improvement (*p* = 0.013, coef: −1.66, CI 95% (−2.99 to −0.35) - BMI: higher BMI (*p* < 0.001) → greater knee ROM improvement (coef: −0.25, CI 95% (−0.38 to −0.13)

SD: Standard Deviation; BMI: Body Mass Index; m: months; N: number; y: year; ♀: female; ♂: male; TKA: total knee arthroplasty; ♀: female; ♂: male; WOMAC: Western Ontario and McMaster Universities Arthritis Index; MOS: Medical Outcomes Study; HUI3: Health Utility Index 3; FU: follow-up; KSS: Knee Society Score; OKS: Oxford Knee Score; MCS: mental composite score; PCS: physical composite score; KL: Kellgren-Lawrence grading system; ROM: range of motion; SF-12: Short Form 12; OA: osteoarthritis; KOOS PS: Knee injury and Osteoarthritis Outcome Score Physical Function Short Form; SF-36 PCS: Short Form 36 physical composite score; CCI: Charlson Comorbidity Index; SF-12 PCS: Short Form 12 physical composite score; SF-12 MCS: Short Form 12 mental composite score; TENS: transcutaneous electrical nerve stimulation; KOOS ADL: Knee injury and Osteoarthritis Outcome Score Activities of Daily Living; PPT: pain pressure threshold; QoL: quality of life; NHP: Nottingham Health Profile; CIRS: Cumulative Illness Rating Scale; MetS: Metabolic Syndrome; ACR: American College of Radiology; EQ-5D-3L: EuroQol Five Dimensions Health Questionnaire; miRNA: microRNA; TUG: Timed Up and Go-test; PASS: Patient Acceptable Symptom State; ASA: American Society of Anaesthesiologists; ACR: American College of Rheumatology; KAM: knee adduction moment; KFM: knee flexion moment; IQR: interquartile range; MRI: magnetic resonance imaging; HRQoL: Health-related quality of life; SCT: stair climbing test; 6MWT: 6 min walking test; FJS-12: Forgotten Joint Score 12; KOS-ADLS: Knee Outcome Survey Activities of Daily Living Scale

**Table 3 ijerph-20-05796-t003:** Risk of bias assessment with QUIPS and levels of evidence and conclusion with EBRO.

QUIPS	1	2	3	4	5	6	Overall	Level of Evidence
Amusat et al. (2014) [24]	High	Low	High	Low	Low	Low	**High**	B
Ayers et al. (2022) [63]	Low	High	High	Low	Low	Low	**High**	B
Bin Abd Razak et al. (2016) [25]	High	High	Low	Low	Low	Moderate	**High**	B
Bonnefoy-Mazure et al. (2017) [26]	High	Low	Moderate	Low	Low	Moderate	**High**	B
Çankaya et al. (2016) [27]	Low	Low	High	Low	High	Low	**High**	B
Christensen et al. (2020) [61]	Low	High	Moderate	Low	Moderate	Low	**High**	B
Clement et al. (2013) [28]	High	Low	Moderate	Low	High	Low	**High**	B
Collins et al. (2017) [29]	Low	Moderate	Moderate	Low	Low	Moderate	**Moderate**	B
Collins et al. (2012) [30]	High	High	High	Low	High	Moderate	**High**	B
Cooper et al. (2017) [22]	Low	Moderate	Moderate	Low	High	Moderate	**High**	B
De Leeuw et al. (1998) [31]	High	Moderate	Low	Low	High	Moderate	**High**	B
Deshmukh et al. (2002) [32]	Moderate	High	Moderate	Low	Low	Moderate	**High**	B
Dettoni et al. (2018) [33]	High	Moderate	Low	Low	Low	Moderate	**High**	B
Gandhi et al. (2010) [34]	Low	Low	Moderate	Low	Low	Moderate	**Moderate**	B
Gandhi et al. (2010) [35]	Low	Moderate	High	Low	Low	Moderate	**High**	B
Gandhi et al. (2013) [36]	Low	Moderate	Low	Low	Low	Moderate	**Moderate**	B
Giesinger et al. (2018) [37]	High	High	High	Low	High	Low	**High**	B
Giordano et al. (2020) [38]	Moderate	Low	Low	Low	Low	Moderate	**Moderate**	B
Hakim et al. (2020) [39]	Low	High	Moderate	Low	High	Low	**High**	B
Hodges et al. (2018) [62]	Low	Low	Low	Low	Low	Moderate	**Low**	A2
Järvenpää et al. (2012) [40]	Low	Moderate	High	Low	High	Moderate	**High**	B
Jauregui et al. (2016) [41]	Moderate	Moderate	Moderate	Low	Moderate	Moderate	**Moderate**	B
King et al. (2021) [64]	Low	High	High	Low	High	Moderate	**High**	B
Lamb et al. (2003) [42]	High	Low	Low	Low	Low	Moderate	**High**	B
Lampe et al. (2016) [20]	Low	High	High	Low	Low	Moderate	**High**	B
Lampe et al. (2016) [21]	Low	Low	Moderate	Low	Low	Moderate	**Moderate**	B
Li et al. (2017) [43]	Low	High	High	Low	Low	Moderate	**High**	B
Lizaur-Utrilla et al. (2014) [44]	High	Low	Moderate	Low	Moderate	Moderate	**High**	B
McQueen et al. (2007) [45]	High	Low	Moderate	Low	High	Low	**High**	B
Merle-Vincent et al. (2011) [46]	Moderate	Moderate	Moderate	Low	Low	Moderate	**Moderate**	B
Mishra et al. (2022) [65]	High	High	High	High	High	Moderate	**High**	B
Núñez et al. (2007) [47]	Low	High	Moderate	Low	Low	Low	**High**	B
Núñez et al. (2011) [23]	Low	Moderate	Low	Low	Moderate	Moderate	**Moderate**	B
Overgaard et al. (2019) [48]	Low	Low	High	Low	Low	Moderate	**High**	B
Paterson et al. (2020) [49]	Low	High	Moderate	Low	Low	Moderate	**High**	B
Paxton et al. (2016) [50]	Low	High	High	Low	Low	Moderate	**High**	B
Petersen et al. (2020) [51]	Low	High	Moderate	Low	High	Moderate	**High**	B
Pua et al. (2019) [52]	Low	Low	Moderate	Low	Moderate	Moderate	**Moderate**	B
Rissolio et al. (2021) [66]	Moderate	High	High	Low	High	Moderate	**High**	B
Scott et al. (2016) [53]	High	Low	High	Low	Moderate	Moderate	**High**	B
Sideris et al. (2022) [67]	Low	Moderate	Low	Low	High	Moderate	**High**	B
Steinhaus et al. (2019) [54]	Moderate	Moderate	High	Low	Moderate	Moderate	**High**	B
Stevens-Lapsley et al. (2010) [55]	Moderate	High	Moderate	Low	High	Low	**High**	B
Sveikata et al. (2017) [56]	Low	Moderate	Moderate	Low	Low	Moderate	**Moderate**	B
Tchetina et al. (2020) [57]	Low	Moderate	Low	Low	High	Moderate	**High**	B
Teo et al. (2018) [58]	Moderate	Moderate	High	Low	High	Moderate	**High**	B
Torres-Claramunt et al. (2016) [59]	Moderate	High	Moderate	Low	High	Moderate	**High**	B
Zeni et al. (2010) [60]	Moderate	High	Moderate	Low	Low	Moderate	**High**	B
Zhang et al. (2021) [68]	Low	High	High	Low	Low	Moderate	**High**	B

1. Study Participation. 2. Study Attrition. 3. Prognostic Factor Measurement. 4. Outcome Measurement. 5. Study Confounding. 6. Statistical Analysis and Reporting.

**Table 4 ijerph-20-05796-t004:** Overview of all outcome measures.

Outcome	Pain	Functional Disabilities				QoL	Satisfaction
		Functional Impairments	Activities/Limitations	Gait Impairments	Gait Activities/Limitations		
**Outcome measures**	WOMAC pain [23,24,29,36,44,45,47,64] KOOS pain [26,43,48] OKS pain [52] VAS [38,51,57,65] NRS [67]	ROM [21,40,52,58] KSS knee [30,32,33,40,41,44,68] WOMAC stiffness [23,45,47] KSS ROM [68]	WOMAC total [23,33,34,35,40,45,56,66,68] WOMAC function [23,24,26,29,44,45,47] KOOS-PF [27,64] KOOS ADL [22,48] OKS [28,37,58] KSS total [33,39,58,59,65] KSS function [21,30,32,33,39,40,41,44,65,68] NHP [32] Active Australia survey [55] KOS-ADLS [60] Sedentary behaviour (hours sitting/24 h) [62] Reported physical activity (minutes/week) [50] Perceived physical function [61] Good outcome * [25]	ROM (knee flexion) during gait cycle [26] Gait biomechanics [49]	Gait velocity [22,26,42] Daily activity (steps/day) [22] Walking distance [40] TUG [40,55] Stair climbing speed [42,55] Time able to walk [52] 6MWT [55,64] Stair climbing task [60]	SF-12 [26,28,44,56] SF-36 [27,34,43,45,55,58,59,68] Rosser Index Matrix [31] EQ-5D-3L [37,54] PROMS [65]	“how satisfied are you with your operated knee?” [30,37,46,53,61,63,66] PASS [64] FJS-12 [66]

WOMAC: Western Ontario and McMaster Universities Arthritis Index; KOOS pain: Knee Osteoarthritis Outcome Score pain; OKS: Oxford Knee Score; VAS: Visual Analogue Scale; NRS: Numeric Rating Scale; ROM: range of motion; KSS: Knee Society Score; KOOS PS: Knee injury and Osteoarthritis Outcome Score Physical Function; KOOS ADL: Knee injury and Osteoarthritis Outcome Score Activities of Daily Living; NHP: Nottingham Health Profile; KOS-ADLS: Knee Outcome Survey Activities of Daily Living Scale; TUG: timed up and go test; 6MWT: 6 min walking test; SF-12: Short Form 12; SF-36: Short Form 36; EQ-5D-3L: EuroQol Five Dimensions Health Questionnaire; PROMS: Patient-reported outcome measures; PASS: Patient Acceptable Symptom State; FJS-12: Forgotten Joint Score * A ‘good outcome’ was defined by a combination of the Oxford Knee Score and the Pain Catastrophizing Scale.

**Table 5 ijerph-20-05796-t005:** Predictive factors.

		Univariate Predictive Factors			Multivariate Predictive Factors		
	Outcome Measure	Predictive in Favour of Obese Patients	Predictive in Favour of Non-Obese Patients	Not Predictive	Predictive in Favour of Obese Patients	Predictive in Favour of Non-obese Patients	Not Predictive
**BMI**	Pain	[1,2]	[3]	[4,5]	[43]	[47]	[29,43,48,52]
	Functional imp	[6]	[7,8]	[9,10]	[68]	[21]	[32,33,41,44,47,52,68]
	Functional act/lim	[11]	[12,13,14,15,16,17,18]	[19,20,21]	[68]	[20,22,35,41,48,50,55,62]	[25,29,32,33,34,44,47,56,61,62,68]
	Gait imp	/	/	[22]	/	/	[26,49]
	Gait act/lim	/	/	[23]	/	[22,42]	[26,42,52,55,60]
	Satisfaction	/	[24]	/	/	[46]	[29,53,61,63]
	QoL	[25,26]	[27,28,29,30]	[31,32,33,34]	[68]	[43,54]	[34,43,55,56,68]
	Outcome measure	Predictive in favour of diabetes	Predictive in favour of no diabetes	Not predictive	Predictive in favour of diabetes	Predictive in favour of no diabetes	Not predictive
**Diabetes**	Pain	/	/	/	/	[24]	[52,64]
	Functional imp	/	/	[35]	/	[52,68]	[41]
	Functional act/lim	/	[36]	[37]	/	[24]	[28,34,41,50,64,68]
	Gait imp	/	/	/	/	/	/
	Gait act/lim	/	/	/	/	/	[52,64]
	Satisfaction	/	/	[38]	/	[64]	/
	QoL	/	/	[39]	[28]	/	[28,68]
	**Outcome Measure**	**Predictive in Favour of Higher Cytokine LEVEL**	**Predictive in Favour of Lower Cytokine Level**	**Not Predictive**	**Predictive in Favour of Higher Cytokine Level**	**Predictive in Favour of Lower Cytokine Level**	**Not Predictive**
**Cytokine Levels**	Pain	[40,41]	/	/	/	[36,57]	[38]
	Functional imp	/	/	/	/	/	/
	Functional act/lim	/	/	/	/	/	/
	Gait imp	/	/	/	/	/	/
	Gait act/lim	/	/	/	/	/	/
	Satisfaction	/	/	/	/	/	/
	QoL	/	/	/	/	/	/
	Outcome measure	Predictive in favour of dyslipidaemia	Predictive in favour of no dyslipidaemia	Not predictive	Predictive in favour of dyslipidaemia	Predictive in favour of no dyslipidaemia	Not predictive
**Dyslipidaemia**	Pain	/	/	/	/	/	[52]
	Functional imp	/	/	/	/	/	[52]
	Functional act/lim	/	/	/	/	/	[35]
	Gait imp	/	/	/	/	/	/
	Gait act/lim	/	/	/	/	/	[52]
	Satisfaction	/	/	/	/	/	/
	QoL	/	/	/	/	/	/

BMI = Body Mass Index; imp = impairments; act = actities; lim = limitations; QoL = Quality of Life.

## Data Availability

Not applicable, no new data were created.

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
