# Peer review of "Identification of Metabolic Factors and Inflammatory Markers Predictive of Outcome after Total Knee Arthroplasty in Patients with Knee Osteoarthritis: A Systematic Review"

_ijerph, 2023, doi:10.3390/ijerph20105796_

Round 1
Reviewer 1 Report
Thank you for the opportunity to review this article. This is a very thorough and well-written review. The authors clearly put forth a lot of effort into the preparation of this review. I include some minor comments below; however, I would be curious to understand if including retrospective studies would add consistency/clarify to some of the heterogeneity of the results? (comment 5 below).
Abstract
1/ “Hard” – Challenging may be a better word
Methods
2/ Very thorough and well-described
3/ 2.6: Risk of Bias: How many people scored the studies? Were the scores consistent? Was there a measure of reliability in the scores?
4/ When examining the merit of the studies – do the ones with greater sample sizes hold more weight? How did you account for differences in sample sizes between the studies? I assume this was implicit from RoB (1) study participation – what was the threshold for low/moderate/high for each of these domains in ROB? For (1) For example, what was the threshold for sample size for each category?
Results and discussion:
5/ Since 38 studies suffered a high RoB, would it be useful to also include retrospective studies in the review? While retrospective studies may not provide information on causation, they do provide important information on associations and would add to the number of studies available (with potentially larger sample sizes in general). Would it be possible to run a sensitivity analysis to include some of these to understand if the results are more consistent? This would be especially useful since the current conclusions (with the prospective studies only) were quite heterogeneous with conflicting evidence in the conclusion, and many had high RoB.
Figures/Tables
6/ Figure 1 is quite blurry.
7/ Table 2 is very thorough. Well done.
8/ Kindly check a few minor details in Table 2: Add units to BMI, consistency with writing (i.e. some columns have T”X” while others do not.
Reviewer 2 Report
- In Table 1, the exclusion criteria state that non-human research is excluded. What does that mean?
- The resolution in Figure 1 is too low, making it difficult to see.
- For Table 2, did the author distinguish between type I and type II diabetes? As BMI seems not applicable for typ1 I related studies. Minor: it would be better to re-orientate the table for ease of reading.
- On page 36, the author should provide clarifications in the Conclusion section, such as specifying what conflicting results mean or what is meant by "results were never in favor of obese patients".
Reviewer 3 Report
The aim of the article "Identification of metabolic factors and inflammatory markers 2 predictive of outcome after total knee arthroplasty in patients 3 with knee osteoarthritis: a systematic review" was "To identify metabolic factors and inflammatory markers that are predictive of postoperative total knee arthroplasty (TKA) outcome". However, the findings do not reflect any factors that could be considered as predictors. Both the scientific novelty of the study and its practical relevance are unclear. Most of the tables are presented in an entirely unreadable format, and the text is also overloaded with literary references. There are also a number of less significant comments on the manuscript, which, taken together, do not allow it to be recommended for publication. The manuscript not only needs a radical redesign, but also a revision of the research design and/or materials in order to achieve the defined results and research objectives.
Round 2
Reviewer 3 Report
The authors have worked on improving the manuscript, which has improved the perception of the material. I have no further comments